# Chitosan Microparticles Coupled with MAGE-AX and CpGs as a Treatment for Murine Melanoma

**DOI:** 10.3390/pharmaceutics17070932

**Published:** 2025-07-19

**Authors:** Gabriela Piñón-Zárate, Beatriz Hernández-Téllez, Ariel Ramírez-Cortés, Katia Jarquín-Yáñez, Enrique A. Sampedro-Carrillo, Miguel A. Herrera-Enríquez, Christian A. Cárdenas-Monroy, Andrés E. Castell-Rodríguez

**Affiliations:** 1Laboratory of Immunotherapy and Tissue Engineering, Faculty of Medicine, National Autonomous University of Mexico, Mexico City 04510, Mexico; gabrielapinon@unam.mx (G.P.-Z.); beatrizhdezt@facmed.unam.mx (B.H.-T.); ariel.ram.cor@iztacala.unam.mx (A.R.-C.); jy.katy@facmed.unam.mx (K.J.-Y.); sampedro@unam.mx (E.A.S.-C.); mikeh@unam.mx (M.A.H.-E.); 2Laboratory of Genetics, National School of Forensic Sciences, National Autonomous University of Mexico, Mexico City 04510, Mexico; ccardenas@enacif.unam.mx

**Keywords:** chitosan microparticles, immunotherapy, biomaterials, melanoma

## Abstract

**Background/Objectives:** One current cancer treatment is immunotherapy, in which tumor antigens (such as MAGE) or adjuvants (such as CpGs) can be used to induce the destruction of tumor cells by the immune system; however, the therapeutic response is generally weak. Therefore, it is necessary to develop a strategy that increases the immune response induced by tumor antigens and CpGs. We propose the coupling of tumor antigens and adjuvants to chitosan (Cs) microparticles to improve the immune response against cancer, as these microparticles can activate the innate immune response when recognized by macrophages and dendritic cells (DCs). **Methods:** Cs microparticles coupled with CpGs and tumor antigens were constructed with the emulsification method; then, their morphology, in vitro biological effect on DCs, and therapeutic effect in a murine melanoma model were analyzed. **Results:** The Cs microparticles showed a rounded morphology and a size of approximately 5 μ; in addition, they were not cytotoxic in in vitro assays and induced the production of IFNα. Finally, in the murine model of melanoma, treatment with Cs microparticles coupled to MAGE or CpGs reduced the tumor growth rate and increased both survival and the presence of cell death areas in the tumor parenchyma in contrast to the control group. **Conclusions:** The results suggest that treatment with Cs microparticles coupled to tumor antigen and/or CpGs can be considered a promising strategy in the field of immunotherapy based on the use of biomaterials.

## 1. Introduction

Antitumor immunotherapy is based on the use of molecules and cells that are able to induce immune responses, leading to the elimination of tumors. In immunotherapy, it is common to promote the activation of antigen-presenting cells such as DCs and effector cells such as T lymphocytes and NK cells, as these are the most successful cells for developing memory immune responses [1]. Various cytokines or immunomodulatory molecules have been used to induce the activation of DCs and T lymphocytes; however, the short half-life of these molecules in the body often requires the use of high concentrations of biomolecules for an effect to be observable, which can often lead to adverse effects [1,2]. Other alternatives have been developed to increase the half-life of the molecules used in classical immunotherapy, allowing them to be effectively recognized by other cells of the immune system [3].

Currently, scaffolds and microparticles are constructed from biomaterials so that they may be recognized by leukocytes, thereby promoting their activation and migration. The biomaterials used can facilitate the prolonged release of conjugated bioactive compounds, such as immunomodulatory molecules or antigens, and help extend their half-life. Microparticles can be transported through the bloodstream, release their content into the extracellular medium, and be recognized by leukocytes. They can also be manufactured from multiple materials, such as metals, ceramics, or biopolymers, such as chitosan (Cs) [4]. Cs is a semi-synthetic, non-toxic, and biocompatible N-acetyl copolymer derived from the N-acetylation of chitin from fungi or crustaceans. It has been widely used to fabricate microparticles, hydrogels, or microspheres conjugated with adjuvants, antibodies, and antigens, or even loaded with chemotherapeutic agents [5]. Furthermore, Cs can be recognized by antigen-presenting cells of the innate immune system via TLR4 and cGAS/STING, which promotes the maturation as well as activation of these cell types and the production of type I IFN [6]. Thus, Cs microparticles exacerbate the effect induced by poorly immunogenic antigens, increasing the production of Th1 cytokines, IL-2, and IFNγ [4].

Microparticles can be coupled with other immunogenic molecules, such as TLR ligands and tumor antigens, to increase and target the immune response induced by Cs [4]. Unmethylated cytosine–phosphate–guanosine ODNs (CpGs) are synthetic unmethylated cytosine and guanine dinucleotides recognized by TLR9 that mimic the presence of viral or microbial DNA in a tissue, inducing the activation of macrophages and DCs [7]. CpGs have been used as adjuvants to increase the immunogenicity of the antigens used in antitumor immunotherapy, since they induce the production of IL-12 as well as IFNα and stimulate NK cells as well as plasmacytoid DCs [8]. These cytokines are crucial to the development of a Th1 response, which is involved in the activation of CD8 T lymphocytes and NK cells necessary for the elimination of tumor cells [9]. On the other hand, tumor antigens help ensure that immune responses are specific [10]. Several antigens have been used in the treatment of melanoma; nevertheless, MAGE antigens have been the most used, as they can induce the development of a Th1 immune response dependent on CD8 T lymphocytes [11].

Thus, we aimed to investigate if conjugating tumor antigens such as MAGE-AX and CpGs to microparticles results in a reduction in tumor growth and growth rates when inoculated in mice with induced melanoma. Additionally, a histopathological analysis of the tumor parenchyma was conducted to analyze the therapeutic effectiveness of Cs microparticles coupled with MAGE-AX and CpGs.

## 2. Materials and Methods

### 2.1. Mice

For the in vivo experiments, for each experimental group, six male C57BL/6 mice with H2Kb (RRID:MGI:5577054) from six to eight weeks of age were used. Mice were kept under controlled light–darkness and temperature conditions and fed ad libitum in the Bioterium of the Department of Cell and Tissue Biology of Facultad de Medicina, UNAM. A maximum of four mice were kept in boxes with solid and continuous floors, walls with removable grille lids with rounded edges, and sawdust beds in clean and dry conditions. Food and drinking water were freely accessible. All the above measures were in accordance with the Official Mexican Standard of technical specifications for the production, care, and use of laboratory animals (NOM-062-ZOO-1999).

### 2.2. Reagents

The following reagents were used: 75% deacetylated crab chitosan ≥ (poly(D-glucosamine) deacetylated chitin) of high viscosity with low molecular weight and water insoluble (Sigma-Aldrich, St. Louis, MO, USA); 25.19% *w*/*w* glutaraldehyde; mineral oil of variable Mw (205 and 500 g/mol), with a boiling point from 260 °C to 360 °C and a specific density of 0.845 to 0.905 for heavy oil and 0.818 to 0.880 for light oil (Farmacia París, Mexico City, Mexico); albumin fraction V; SPAN 80 (Sigma-Aldrich, St. Louis, MO, USA); and glacial acetic acid at 17.48 M in addition to hexane at 86.18 g/mol (Sigma-Aldrich, St. Louis, MO, USA).

### 2.3. Cs Microparticle Manufacture

The Cs microparticles were prepared following the methodology used by Jarquín-Yáñez, 2017 [12]. Briefly, an aqueous solution was prepared by mixing 10 mL of distilled water, 200 µm of acetic acid, 0.2 g of chitosan and 0.01 g of albumin. The solution was kept under agitation overnight. Subsequently, the solution was incorporated into a 10 mL syringe for dropwise addition to the next solution. An oily solution consisting of 99 mL of mineral oil and 1 mL of SPAN 80 was manufactured and was stirred for 10 min at 750 rpm. The oily solution was then homogenized with an Ultraturrax T25 (IKA Works, Inc., Wilmington, NC, USA) at 10,000 rpm, while the aqueous solution was incorporated into the oily one with dropwise addition. Afterwards, the homogenization speed was raised to 12,000 rpm for one minute. The resulting emulsion was kept under stirring at 750 rpm for 30 min; then, 10 mL of 25% glutaraldehyde was added to the emulsion drop-by-drop, and the total solution was stirred at 750 rpm for 2 h. Finally, the emulsion was centrifuged at 4000 rpm for 10 min at 4 °C, and the microparticles were washed first with hexane and then with distillate water. Afterwards, the microparticles were dried, weighed, and then incubated with 25 mM MAGE-AX (Invitrogen, Leiden, The Netherlands) and/or the ODN 2395 TLR9 ligand (CpGs) (Invivogen, San Diego, CA, USA) for 12 h. The size of the microparticles was then verified with flow cytometry. Finally, 1–5 µm glass microparticles were employed as a size control.

### 2.4. Processing for Scanning Electron Microscopy (SEM)

Once the microparticles were obtained, they were dissolved in distilled water, and 10 µg was placed on a barrel sample holder. The sample was left to dry overnight; then, it was subjected to a gold bath for 2 s. Finally, the microparticles were observed with a high-vacuum scanning microscope.

### 2.5. Bone Marrow-Derived Dendritic Cell Differentiation and Treatment with Cs Microparticles

Bone marrow dendritic cells (bmDCs) were seeded with the microparticles to analyze the latter’s biological effect. Briefly, the C57BL/6 mice were euthanized with the administration of a lethal dose of sodium pentobarbital. Subsequently, the tibias and femurs were obtained and washed with 70% ethanol and then with Hank’s Balanced Salt Solution (HBSS). Next, the epiphyses were removed from the bones to be perfused with RPMI-1640 medium to obtain a solution rich in bone marrow precursor cells. Then, 5 × 10^5^ cells/mL were incubated in RPMI 1640 medium (Gibco, Thermo Fisher Scientific, San Francisco, CA, USA) supplemented with 10% Fetal Bovine Serum, 1% of an antibiotic/antifungal cocktail, and 20 ng/mL of GM-CSF (Biolegend, San Diego, CA, USA) at 37 °C and 5% CO_2_. The cells were collected on day six of culture.

In order to determine the effect of the microparticles, the bmDCs were seeded with 80, 160, 380, or 760 μg of Cs microparticles for 24 h in RPMI-1640 medium supplemented with SBF (Gibco, Thermo Fisher Scientific, San Francisco, CA, USA) and antibiotics/antifungals (Sigma-Aldrich, St. Louis, MO, USA). For a a total of 5 h before completing the 24 h treatment, the cells were treated with 5 μg/mL of Brefeldin (Biolegend^®^, San Diego, CA, USA). Subsequently, the cells were stained with anti-IL-12 (RRID:AB_315373) or anti-IFNα coupled with PE (Biolegend^®^, San Diego, CA, USA) and analyzed with flow cytometry and the Flow Jo 10.10 software.

### 2.6. Cytotoxicity Test with Calcein Staining and Ethidium Homodimer

Murine splenocytes were cultured with 80, 160, 380, and 760 µg/mL of microparticles for 48 h, and the life and death assay (Thermo Fisher Scientific, San Francisco, CA, USA) was performed to analyze the microparticles’ cytotoxic effect. The cells were stained with a solution containing 0.5 μL of calcein and 2 μL of ethidium homodimer. As a positive control for cell death, cells were treated with 500 µL of ethanol for an hour. Afterwards, the samples were analyzed with a Nikon Eclipse 80i (Tokyo, Japan) fluorescence microscope, 200× photomicrographs were taken from different fields of each treatment, and the percentage of viable cells was analyzed by using Image-J 1.54 software.

### 2.7. Melanoma Cell Line and MAGE-AX Antigen

The murine melanoma cell line B16-F10 with haplotype H-2Kb (RRID:CVCL_0159) was purchased from the American Type Culture Collection, USA. The MAGE-AX peptide (LGITYDGM) with a purity of 94% was synthesized by Research Genetics (Invitrogen, Leiden, The Netherlands).

### 2.8. Melanoma Induction in B16-F10 Mice

For the induction of melanoma, 60,000 cells of the B16-F10 melanoma line were inoculated subcutaneously into the C57BL/6 mice in the abdominal region.

### 2.9. Tumor Growth Rate and Survival

The C57BL/6 mice were first inoculated once a week for two weeks with 760 µg of Cs microparticles coupled to (A) MAGE-AX, (B) CpGs or (C) MAGE-AX/CpGs. Three weeks later, the mice were challenged with 60,000 melanoma cells. The mice were monitored daily and, once tumor lesions became visible, the largest and smallest diameters were measured every two days by using an electronic vernier. The tumor volume was calculated by using the formula V = (A^2^ × B)/2, where A is the smallest diameter and B is the largest tumor diameter.

### 2.10. Histopathological Evaluation of Tumor Lesions

The tumor lesions were dissected and then fixed for 48 h in 10% formalin for histopathological evaluation. The samples were washed, dehydrated, and embedded in paraffin. Subsequently, sections of approximately 7 µm were made and stained with hematoxylin and eosin (Chemicals Segale, Mexico) for the histopathological study.

### 2.11. Analysis of Results

The log-rank test (Mantel–Cox) was used for the survival analysis of the mice treated with the microparticles. Furthermore, we evaluated the effect of the microparticles on the induction of cytotoxicity and the production of IFN as well as IL-12 with an ANOVA test and then Tukey’s test to verify changes between groups. All the results were analyzed by using GraphPad 9 software, and a *p* < 0.05 was considered significant.

## 3. Results

### 3.1. Size of Cs Microparticles

The microparticles were observed with SEM for the analysis of their shape and surface. It was found that they were rounded with an irregular surface with some smooth sections (Figure 1A). On the other hand, the flow cytometry assays revealed that we obtained microparticles ranging in size from 1 to 5 μm, although approximately 48% of the microparticles were 2 μm in size (Figure 1B).

### 3.2. Cytotoxicity of Cs Microparticles

The splenocytes were cultured with microparticles at doses of 80, 160, 380, and 760 µg/mL for 24 and 48 h for the analysis of the cytotoxicity induced by the microparticles. Live cells were stained with calcein, which induced a green color, while dead cells were stained with ethidium homodimer and were seen in bright red. No notable changes were observed in cell viability in the splenocyte cultures after 48 h of exposure compared to the control group of viable cells; in addition, the percentages of cells positive for calcein in the 80, 160, 380, and 760 µg/mL cultures were always higher than that observed in the group of splenocytes treated with ethanol, the cell death control group (Figure 2).

### 3.3. DC Treatment with Cs Microparticles

After the analysis of the cytotoxicity induced by the microparticles, the production of IL-12 and IFNα was evaluated after 24 h of incubation of bmDCs with microparticles at different concentrations. Different tendencies were observed. A positive trend in IFNα production was observed at all doses, but especially at 760 μg, in comparison with the control and LPS groups, which we considered the positive control of bmDC maturation (* *p* < 0.05). In contrast, when the effect of microparticles on IL-12 production was studied, the doses of 160, 380, and 760 μg mediated the decreased production of IL-12 in comparison with the LPS group (** *p* < 0.001) (Figure 3).

### 3.4. Survival and Tumor Size

After we proved that Cs microparticles can induce the production of IFNα, the in vivo immunomodulatory effect of the microparticles when used as a treatment for murine melanoma was analyzed. For this purpose and to increase the antitumor immune response, the microparticles were coupled with a melanoma tumor antigen (MAGE-AX) and CpGs, molecules known for mediating an antitumor Th1 response. Thus, the C57BL-6 mice were first inoculated with Cs microparticles, and three weeks later they were inoculated with the melanoma cell line. Once the tumor was visible, the lesions were measured with a vernier. Cs microparticle therapy showed statistically significant changes in both tumor growth and survival compared with the untreated group of mice. In terms of tumor growth, it was observed that mice without treatment showed a marked increase in tumor volume, which reached a peak on day 26 (22.38 cm^2^), while tumor development was very similar in the mice treated with the Cs microparticles either alone or conjugated to CpGs, MAGE-AX, or MAGE-AX/CpGs. From day 26 to 32 post-melanoma inoculation, the groups of mice treated with CS microparticles alone or coupled with CpGs, MAGE-AX, or MAGE-AX/CpGs showed very similar tumor development. By the end of the study, the mice with melanoma treated with the Cs microparticles alone or coupled with MAGE-AX/CpGs showed the smallest tumor volume (Figure 4).

In terms of survival, untreated melanoma-bearing mice survived up to day 28 after being inoculated with melanoma cells (Figure 4). The mice treated with Cs microparticles alone exhibited a survival rate of 60% on day 32. The mice treated with Cs microparticles/MAGE-AX/CpGs exhibited a survival rate of approximately 55% up to day 32. The mice treated with Cs microparticles/MAGE-AX and Cs microparticles/CpGs exhibited consistent levels of survival over time, 80% from day 22 to 32. Therefore, the groups treated with Cs microparticles, Cs microparticles/MAGE-AX, and Cs microparticles/CpGs resulted in higher survival rates than those with Cs microparticle/MAGE-AX/CpG treatment and WT.

### 3.5. Histopathological Findings

The examination of different microscopic fields allowed us to identify a sequence of changes during tumor progression. Initially, the histological growth pattern was formed by compact sheets of melanoma cells. Over time, areas with a loss of cohesiveness were observed, which formed elongated holes or sinuses with irregular contours that were eventually filled with proteinaceous material. The sinuses increased in number and size and showed an inflammatory infiltrate mainly consisting of lymphocytes with few polymorphonuclear neutrophils. Tumor cells adjacent to the infiltrate showed gradual degenerative changes, and, subsequently, large necrotic areas that seemed to spare the peripheral tumor areas near blood vessels were formed, generating a characteristic appearance of pseudo-rosettes. Later, the necrotic areas increased in size and eventually occupied large areas of the tumor (Figure 5).

The histopathological changes described were observed in samples with varying degrees of progression and extension in each of the groups studied. In the group treated with Cs microparticles, the tumor size did not show a significant decrease, and necrotic changes affected between 10 and 20% of the tumor (Figure 5A,F). In the Cs microparticle/CpG group, a significant decrease in tumor size and necrosis affecting between 30 and 40% of the tumor was observed (Figure 5B,F). For the Cs microparticle/MAGE-AX group, there was a marked decrease in tumor size, with necrosis extending to between 40 and 50% of each tumor (Figure 5C,F). Finally, the Cs microparticle/CpG/MAGE-AX group showed the smallest overall tumor sizes and extension of the necrosis process, which varied between 50 and 70% of the tumor mass (Figure 5D,F).

## 4. Discussion

It is well known that there are multiple types of immunotherapies aimed at the activation of the immune system, such as adoptive therapy and the use of cytokines or antibodies [13], which have helped to increase the survival of cancer patients [14]. These therapies work based on the activation of CD4, CD8, and NK T lymphocytes, which migrate to the tumor parenchyma to promote tumor cell death and slow down tumor growth [15]. However, there are patients who do not respond to therapy, especially due to the inhibitory microenvironment in the tumor stroma, developing tolerogenic immune responses that allow for the proliferation of tumor cells [15]. Therefore, additional strategies are needed to enhance the immune response generated by antitumor immunotherapy in the tumor parenchyma. Microparticles coupled with immunomodulatory molecules are an option, since they can be administered near or in the tumor lesion, which helps modulate the antitumor immune response on account of the interaction with DCs or the release of coupled molecules such as tumor antigens, cytokines, and TLR ligands [16]. The use of microparticles becomes relevant because, in addition to the above, it preserves the molecules attached to them by preventing their prompt degradation. Thus, we propose the use of Cs microparticles as an antitumor treatment strategy, since Cs is a material that is recognized by DCs, which are relevant cells in the induction of an adaptive immune response. Additionally, we investigated the possibility of making this response both specific, by attaching a melanoma antigen, MAGE-AX, to the microparticles, and of greater magnitude, by also coupling CpGs. Thus, the addition of MAGE-AX and CpGs to the microparticles would increase the success in developing an antitumor immune response, making the microparticles an excellent tool for cancer treatment.

In this study, Cs microparticles were constructed and their morphology observed with a scanning electron microscope. It was confirmed that the microparticles showed an approximate size of 1 to 5 microns, which makes them excellent candidates for therapy administration, as they can travel through the circulation or lymphatic vessels and interact with antigen-presenting cells such as DCs or macrophages in secondary lymphoid organs and in the skin [17]. This is relevant since it has been confirmed that Cs can be recognized by DCs and promote cross-presentation. For instance, a study on manufactured Cs/calcium phosphate nanosheets coupled with antigens confirmed that DCs could phagocytose chitosan, process it, and promote the maturation of DCs as well as the production of Th1 cytokines [18]. Therefore, given the size of the synthesized microparticles, it is very likely that successful antitumor immune responses could be induced.

Despite the above, it is important to determine whether the synthesized microparticles could have any cytotoxic effect. Therefore, in vitro tests were performed to determine the cytotoxic effect of the microparticles on murine splenocyte cultures. The microparticles were manufactured by using a different method than that for other types of chitosan microparticles, and it was confirmed that they did not induce a decrease in the percentage of viable splenocytes (Figure 2). It is noteworthy that the properties of Cs have already been widely studied, especially biocompatibility, biodegradability, and cytotoxicity, given that it is a biomaterial used in food, cosmetics, and the treatment of skin wounds [19]. The microparticles we synthesized were observed to induce zero cytotoxicity, which makes them excellent candidates to be used as adjuvants, carriers, and releasers of molecules that can induce antitumor immune responses. We next aimed to establish the immunomodulatory effect of the Cs microparticles coupled with MAGE-AX and CpGs to determine whether they could be used in antitumor immunotherapy.

The effect of Cs microparticles administered at different concentrations in murine splenocytes was analyzed to further determine their immunomodulatory effect. A trend was evident: the 760 μg dose induced the greatest production of IFNα (Figure 3). IFNα is known to trigger immunological responses against viruses and is produced when pathogen-associated molecular patterns are recognized by RIG I-like receptors located on DCs and monocytes [20]. This suggests that these cells recognized the Cs microparticles, especially due to their negative charge, which helps biomaterial to be easily recognized by cells, in addition to binding to multiple molecules [5]. Given its immunomodulatory characteristics, Cs has been used as an adjuvant in vaccines against Newcastle disease [21], parvovirus [22], and salmonella [23], promoting IFNγ production and the proliferation of CD4 as well as CD8 T lymphocytes [24,25,26]. Therefore, not only does Cs help to protect and deliver the molecules it attaches to, but it may also have adjuvant properties. Thus, it has been proven that Cs particles can induce DC maturation by increasing MHCII and CD86 expression by activating the cGAS-STING pathway as well as the production of IFNα, and inducing an increase in T CD4 and T CD8 lymphocyte levels [6,24]. An enhanced humoral immune response has also been shown with the coupling of Cs microparticles to the anthrax vaccine [27]. Therefore, the Cs microparticles constructed by using the emulsification method could be recognized by DCs or monocytes, inducing the production of IFNα. The above is relevant, since it has been proven that the production of IFNα helps the activation of cytotoxic cells such as natural killer cells, in addition to the increase in the expression of class I MHC molecules, which promotes the recognition of tumor cells by CD8 T lymphocytes [28,29], which are all essential elements in the antitumor immune response needed to reduce tumor size and improve survival [30]. Therefore, the fact that Cs microparticles constructed by the emulsion method promoted the above confirms their merits for use in antitumor immunotherapy.

Therefore, the Cs microparticles were used to induce an effective immunological response against tumor lesions, specifically melanoma, a very aggressive type of cancer [31]. CpGs and a tumor antigen, MAGE-AX, were coupled with the Cs microparticles to enhance the induced immune response and increase its specificity, especially because they are immunomodulatory molecules that help develop a Th1 response [32,33].

Given the above, a melanoma murine model was used to determine the therapeutic properties of the microparticles coupled with CpGs and MAGE-AX. Based on our previous trials, which indicated that a dose of 760 µg of Cs microparticles could promote the production of IFNα, this concentration was used for the treatment of the mice. The immunomodulatory properties of the microparticles alone were also observed, since they were also used without being coupled with CpGs and/or MAGE-AX. In line with several investigations that confirm the properties of Cs, treatment with Cs-only microparticles was able to induce an effective tumor response, as evidenced by the treated mice showing an enhanced survival rate compared with the control group. Cs microparticles have already been used as carriers of immunomodulatory molecules, in addition to being used as an adjuvant in vaccines against multiple infectious agents [26]. In the case of antitumor immunotherapy, Cs has been used in the form of hydrogels and microparticles. For example, Cs microparticles have been coupled with raloxifene, which promoted the death of A549 cells [34], and Cs microparticles coupled with cisplatin have also been used [35]. Additionally, Cs has been shown to induce the polarization of M1 macrophages and DCs, cells that can activate T and NK lymphocytes, which are necessary for tumor cell death [36]. We also found that Cs microparticles could activate DCs in vitro, so it is possible that their inoculation could induce the activation of DCs in vivo. Therefore, Cs particles can be considered an excellent immunotherapeutic strategy due to their adjuvant properties and role as molecule carriers. Subsequently, the effect of microparticles coupled with MAGE-AX and/or CpGs was examined. We observed similar results in terms of volume and survival rates, as well as the appearance of areas of cell death, compared to the groups treated with microparticles containing only Cs or Cs/MAGE-AX/CpGs. Only the group of mice with melanoma treated with Cs/MAGE-AX/CpGs demonstrated greater survival and tumor volume than the mice treated with Cs microparticles. This indicates that the addition of MAGE-AX or CpGs increased the immunogenicity of the Cs microparticles. CpGs are recognized by (TLR9), which can induce DC activation [8], while MAGE-AX is known to induce a CD8 T lymphocyte-dependent response when used alone [37] or when bone marrow-derived DCs are loaded with the peptide [11]. It is noteworthy that the survival observed when Cs microparticles are used is superior to that found when the peptide is used with DCs or when both MAGE-AX and CpGs are coupled with GE/HA scaffolds [11,33]. When MAGE-AX and CpGs were combined with the Cs microparticles, the response was not greater than that observed when the molecules were separately coupled with the microparticles. All the above is relevant, since it is possible that the loading amounts of CpGs and MAGE-AX were different when these molecules were coupled with the microparticles individually and together, with the latter case possibly inducing their doses to decrease in the Cs microparticles. Further release studies will be necessary to confirm this hypothesis.

## 5. Conclusions

In this study, Cs microparticles were constructed. The ultrastructure and size of the Cs microparticles were verified, and in vitro tests with murine splenocytes confirmed that the Cs microparticles did not induce cytotoxicity and had immunomodulatory properties, as evidenced by the promotion of the production of IFNα. MAGE-AX and CpGs were coupled with the Cs microparticles for treatment against murine melanoma to determine their in vivo immunomodulatory properties. We found that the Cs microparticles promoted the development of an antitumor immune response, which was reflected in the appearance of considerable areas of cell death in the tumor parenchyma, the decrease in the tumor growth rate, and the increase in the survival rate in all experimental groups, but especially in the groups treated with Cs microparticles/CpGs and Cs microparticles/MAGE-AX. These are important results showing that Cs microparticles possess immunomodulatory properties and attesting to their possible efficacy in the development of successful antitumor responses if used in cancer immunotherapy.

## Figures and Tables

**Figure 1 pharmaceutics-17-00932-f001:**
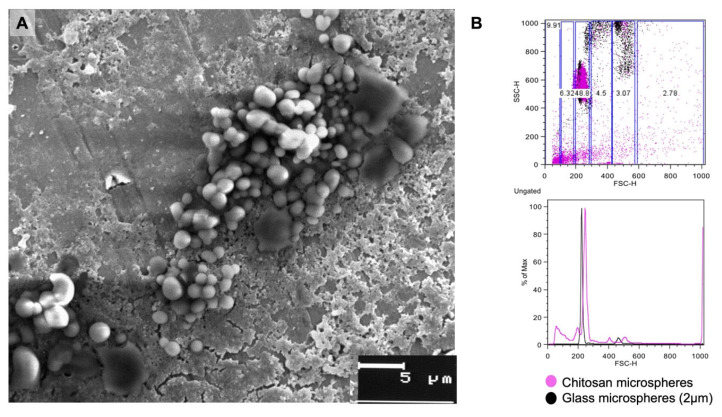
Size of Cs microparticles. The microparticles were observed with SEM, and their size was determined with flow cytometry. (**A**) The microparticles had a rounded shape and an irregular surface. (**B**) The flow cytometry studies determined that there were different particle sizes between 1 and 5 μm, although the microparticles measuring approximately 2 μm were the most abundant.

**Figure 2 pharmaceutics-17-00932-f002:**
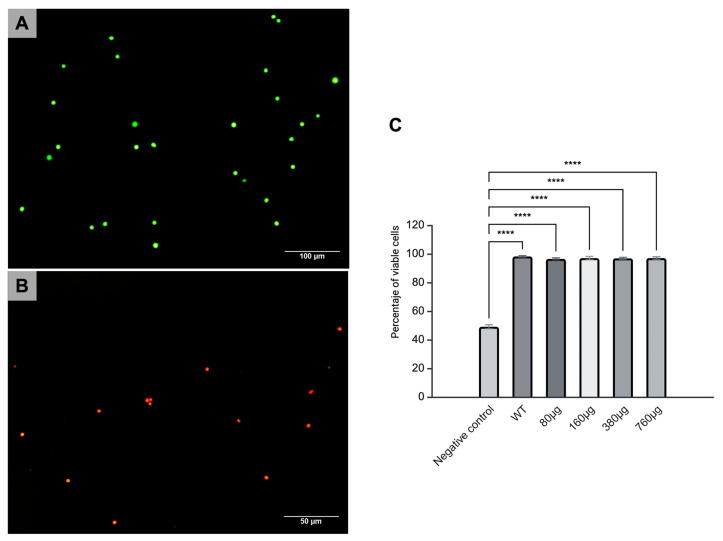
Cytotoxicity of Cs microparticles at different doses in murine splenocytes. Murine splenocytes were incubated for 48 h with microparticles at different doses; then, the presence of viable and non-viable splenocytes was analyzed by using the calcein and ethidium homodimer assay. (**A**) Representative photomicrograph of splenocytes cultured with Cs microparticles and stained with calcein and ethidium homodimer. Cells stained in bright green indicate that cells are viable. (**B**) Representative photomicrograph of positive control for cell death. The cells were incubated for one hour with ethanol. Cells stained in red, indicating cell death, are observed. (**C**) Statistical analysis. The Cs microparticles at different concentrations showed higher cell viability than the cell death control group.: WT vs. 80 µg, 160 µg, 380 µg, and 760 µg; **** *p* < 0.0001: cell death vs. all groups.

**Figure 3 pharmaceutics-17-00932-f003:**
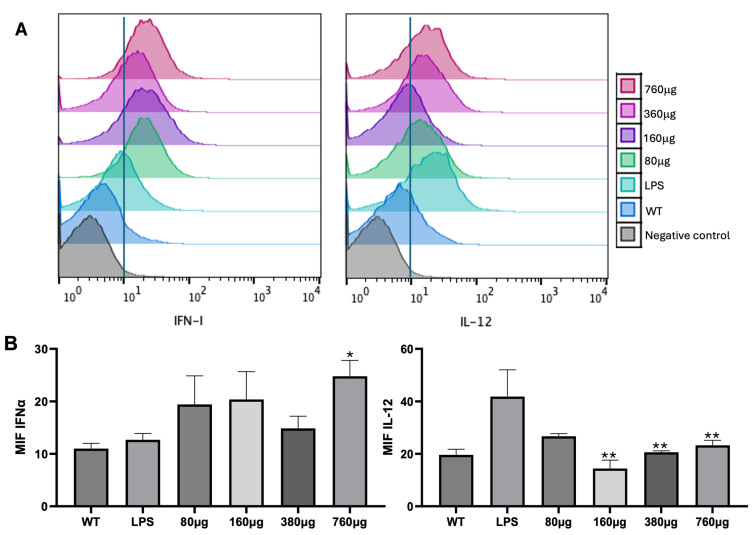
Production of IFNα and IL-12 induced by Cs microparticles in bmDCs. (**A**) Representative histograms of mean intensity fluorescence (MIF) corresponding to the production of IL-12 and IFNα by bmDCs. (**B**) Bar graphs of IL-12 and IFNα MIFs of bmDCs. DCs treated with Cs microparticles promote the production of IFNα, but not IL-12. * *p* < 0.05: IFNα (760 μ vs. WT and LPS); ** *p* < 0.001: IL-12 (160 μ, 380 μ, and 160 μ vs. LPS).

**Figure 4 pharmaceutics-17-00932-f004:**
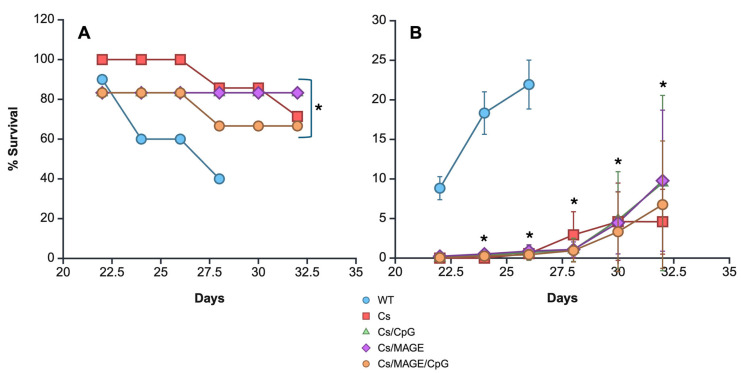
Survival and tumor growth rate of mice inoculated with Cs microparticles and then challenged with murine melanoma cells. (**A**) Survival of mice with melanoma. (**B**) Tumor volume following different treatments. * *p* < 0.05: all groups vs. WT in terms of survival and tumor growth.

**Figure 5 pharmaceutics-17-00932-f005:**
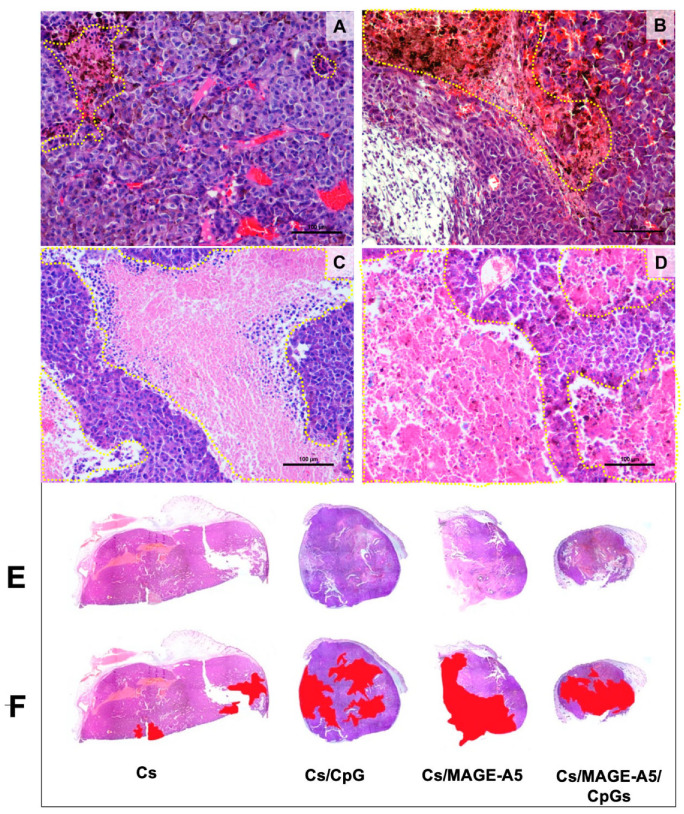
Photomicrographs of tumor parenchyma stained with H and E from mice treated with Cs microparticles coupled to CpGs, MAGE, or both molecules. (**A**) Cs microparticles. (**B**) Cs microparticles/CpGs. (**C**) Cs microparticles/MAGE. (**D**) Cs microparticles/MAGE/CpGs. (**A**) In the tumor parenchyma of mice treated with Cs microparticles alone, multiple capillaries and large areas of basophilic epithelioid tumor cells with euchromatin-rich nuclei are observed. (**B**) In contrast, the parenchyma of mice treated with Cs microparticles/CpGs showed both areas of viable tumor cells and areas with cell death (outlined by yellow dotted lines). (**C**,**D**) The treatments with Cs microparticles/MAGE-AX and Cs microparticles/MAGE/CpGs were those that induced greater areas of cell death in the tumor parenchyma (areas delimited by yellow dotted lines). Scale bar: 100 µm. (**E**,**F**) Holoptic histological images of tumor lesions in mice treated with Cs microparticles coupled with CpGs, MAGE-AX, or both molecules. (**E**) Images of the histological samples of the tumor lesions. Acidophilic regions corresponding to areas of cell death and basophilic regions related to the presence of viable tumor parenchyma are observed. (**F**) Delimitation of the areas of cell death in the tumor parenchyma. The area of cell death in the tumor lesions of mice treated with Cs microparticles/CpGs, Cs microparticles/MAGE-AX, or Cs microparticles/MAGE/CpGs stands out.

## Data Availability

The data supporting the findings of this study are available within the paper, and the datasets generated during and/or analyzed during the current study are available from the corresponding author upon reasonable request.

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
