# Peer review of "Chitosan Microparticles Coupled with MAGE-AX and CpGs as a Treatment for Murine Melanoma"

_pharmaceutics, 2025, doi:10.3390/pharmaceutics17070932_

Round 1

Reviewer 1 Report

Comments and Suggestions for Authors

This manuscript introduces a treatment using chitosan (Cs) microparticles as a delivery system for tumor-associated antigens (MAGE-AX) and Unmethylated cytosine-phosphate-guanosine ODNs (CpGs) using the emulsification method and evaluates their immunomodulatory effects both in vitro and in vivo using a murine melanoma model. While the research is intriguing, several critical issues must be addressed before it can be considered for publication in this journal. Below are the main points for consideration:

Major comments:

  • The manuscript requires thorough proofreading to address grammatical errors and improve readability.
  • What are advantages or the reason behind this research focus on the use of Cs microparticles over Cs nanoparticles?
  • Figure 2 and Section 3.2 need clarification. Figure 2c indicates that Cs microparticles exhibited 100% cell death while the caption suggests that Cs microparticles showed higher cell viability than the cell death control group. How does the author define cell death or death cells group? The discussion in section 3.2 is also confusing.
  • Figure 3a, legend indicate different color profiles is required. Figure 3b, Why the author compared the production of IFN-alpha with WT and LPS group, but only compared the production of IL-12 with LPS group? Important of assessing IFN-alpha and IL-12 production should be addressed.
  • In figure 4, the survival percentage of Mice treated with CS/MAGE-AX mentioned in the main text does not appear to be illustrated in the corresponding figure(s). For clarity and consistency, it is recommended to ensure full content described in the data shown.

Minor comments:

  • There appears to be an inconsistency between the formula and the accompanying explanation. The formula refers to "A2", while the description defines only "A".
  • The annotations of “Cs/MAGE-A5” in Figure 5 is not referenced or described in the text. Please provide a brief explanation in the relevant section.
  • The explanation of the term: “Toll-like receptor (TLR)” has already been clearly introduced earlier on, there is no need to repeat this in subsequent sections. Referring to the original mention would be sufficient and would help improve manuscript’s clarity and conciseness.
  • To improve accessibility for readers who may be less familiar with immunological models, it is recommended to briefly explain the role of the LPS group (Section 3.3) when it is first mentioned.

Comments on the Quality of English Language
  • The manuscript requires thorough proofreading to address grammatical errors and improve readability.

Reviewer 2 Report

Comments and Suggestions for Authors

The manuscript describes the preparation and characterization of chitosan microparticles coupled with MAGE-AX and CpGs for treatment of melanoma. The manuscript lacks novelty and does not address a knowledge gap. The use of microparticles for treatment of cancer lacks clinical feasibility compared to nanoparticles. Other major comments are as follows:

1- The manuscript is full of grammatical and syntax errors.

2- Molecular weight of chitosan is not stated.

3- IC50 values were not calculated.

4- Figure 4 is reported without Mean and S.D. values.

5- Discussion is very poor, and does not provide an interpretation for the merits of the microparticles.

Reviewer 3 Report

Comments and Suggestions for Authors

This study by Piñón-Zárate et al. presents a novel immunotherapeutic approach for melanoma treatment using chitosan microparticles as delivery vehicles for tumor antigens and immune adjuvants. The research demonstrates promising preclinical results while contributing to the growing field of biomaterial-based cancer immunotherapy.

The authors address a critical limitation in cancer immunotherapy: the weak therapeutic response typically induced by tumor antigens and adjuvants alone. Their approach leverages chitosan microparticles' inherent immunostimulatory properties, which can activate innate immune responses through recognition by macrophages and dendritic cells via TLR4 and cGAS/STING pathways. The combination strategy incorporates MAGE-AX (a melanoma-associated antigen) for tumor specificity and CpG oligodeoxynucleotides as TLR9 ligands to enhance immune activation.

This rationale aligns with current understanding that chitosan-based nanoparticles can remodel immunosuppressive tumor microenvironments and promote M1 macrophage polarization, creating conditions favorable for anti-tumor immunity. The selection of MAGE antigens is particularly relevant, as these cancer-testis antigens are expressed in various solid tumors while being restricted to immune-privileged sites in healthy tissue.

The study employs a systematic approach with appropriate controls and well-established methodologies:

  • Microparticle characterization: The emulsification method produced particles of 1-5 µm diameter, suitable for cellular uptake by antigen-presenting cells
  • Comprehensive in vitro evaluation: Cytotoxicity assays using calcein/ethidium homodimer staining and dendritic cell activation studies provide solid foundational data
  • Robust in vivo model: The B16-F10 melanoma model in C57BL/6 mice is well-validated for immunotherapy studies
  • Multiple outcome measures: Tumor growth kinetics, survival analysis, and histopathological evaluation provide comprehensive assessment

Major concerns:

Several methodological concerns warrant attention:

  • Limited mechanistic insight: While the study demonstrates efficacy, the precise mechanisms underlying the observed therapeutic effects remain unclear
  • Dosing rationale: The selection of 760 µg microparticle dose lacks comprehensive dose-response optimization
  • Statistical power: Group sizes of six mice may limit statistical robustness, particularly for survival analyses
  • Loading efficiency: The study lacks detailed characterization of MAGE-AX and CpG loading/release kinetics from microparticles

This study presents a promising proof-of-concept for chitosan microparticle-based melanoma immunotherapy. The demonstrated safety profile, immune activation, and therapeutic efficacy support continued development of this platform. While the approach shows merit, optimization of formulation parameters, deeper mechanistic understanding, and expanded preclinical evaluation will be essential before clinical translation. The work contributes valuable insights to the growing field of biomaterial-enhanced cancer immunotherapy, particularly highlighting chitosan's potential as both a delivery vehicle and immune adjuvant.

I propose the publication of the manuscript after considering the major concerns.

Round 2

Reviewer 1 Report

Comments and Suggestions for Authors

The authors have adequately responded to the previously raised questions and concerns.

Reviewer 2 Report

Comments and Suggestions for Authors

The manuscript lacks novelty and does not address a knowledge gap. The use of microparticles for treatment of cancer lacks clinical feasibility compared to nanoparticles. 

Reviewer 3 Report

Comments and Suggestions for Authors

The authors considered the referees' comments and provided adequate responses, modifying the manuscript accordingly. The references are satisfactory and correctly cited.